# Protein Kinase CK2 and Its Potential Role as a Therapeutic Target in Huntington’s Disease

**DOI:** 10.3390/biomedicines10081979

**Published:** 2022-08-15

**Authors:** Angel White, Anna McGlone, Rocio Gomez-Pastor

**Affiliations:** Department of Neuroscience, School of Medicine, University of Minnesota, Minneapolis, MN 55455, USA

**Keywords:** post-translation modifications, CK2, Huntington’s disease, kinase inhibition, HTT phosphorylation

## Abstract

Huntington’s Disease (HD) is a devastating neurodegenerative disorder caused by a CAG trinucleotide repeat expansion in the *HTT* gene, for which no disease modifying therapies are currently available. Much of the recent research has focused on developing therapies to directly lower HTT expression, and while promising, these therapies have presented several challenges regarding administration and efficacy. Another promising therapeutic approach is the modulation of HTT post-translational modifications (PTMs) that are dysregulated in disease and have shown to play a key role in HTT toxicity. Among all PTMs, modulation of HTT phosphorylation has been proposed as an attractive therapeutic option due to the possibility of orally administering specific kinase effectors. One of the kinases described to participate in HTT phosphorylation is Protein Kinase CK2. CK2 has recently emerged as a target for the treatment of several neurological and psychiatric disorders, although its role in HD remains controversial. While pharmacological studies in vitro inhibiting CK2 resulted in reduced HTT phosphorylation and increased toxicity, genetic approaches in mouse models of HD have provided beneficial effects. In this review we discuss potential therapeutic approaches related to the manipulation of HTT-PTMs with special emphasis on the role of CK2 as a therapeutic target in HD.

## 1. Introduction

Huntington’s disease (HD) is an inherited autosomal dominant neurodegenerative disease that affects approximately 1 in 10,000 individuals worldwide. HD is caused by a CAG trinucleotide repeat expansion in exon 1 of the huntingtin gene (*HTT*), resulting in a mutated poly-glutamine (polyQ) expanded HTT protein (mtHTT) [1]. HD manifests with progressive motor dysfunction commonly presenting with involuntary, jerky movements (chorea), hypokinesia, and unintended weight loss in early stages before developing into muscle rigidity, akinesia, difficulty speaking (dysarthria), and difficulty swallowing (dysphagia) at later stages of the disease [2]. In addition to motor symptoms, cognitive and psychiatric disturbances are also prevalent and often precede motor symptoms [3]. Anxiety and depression are most common, and their prevalence is independent of disease stage. Other symptoms dependent upon disease stage include low self-esteem, feelings of guilty, apathy, and suicidal-like behavior [3]. Unfortunately, no disease modifying therapies are available at the moment for this devastating disease [4].

Despite decades of research since the causal mutation of HD was identified by the Huntington’s disease Collaborative research group in 1993 [1], much remains to be elucidated about the molecular mechanisms underlying mtHTT toxicity and neurodegeneration in HD. PolyQ expansion causes mtHTT protein to misfold and aggregate, preferentially affecting GABAergic medium spiny neurons of the striatum, a brain region that controls movement and some forms of cognition [5,6]. Altered mtHTT conformation is reported to reduce normal function of the protein as well as facilitate aberrant protein-protein interactions or subcellular localization, leading to neurotoxicity [7,8]. The actual role of HTT is still unclear, although it has been implicated in numerous cellular processes in the central nervous system, especially, but not limited, to those related to synaptic health. HTT participates in normal presynaptic function by participating in endocytosis and exocytosis of synaptic vesicles [9,10,11]. HTT also participates in axonal transport by forming complexes with kinesin and dynein [12,13,14,15] and contributes to postsynaptic health by assisting in dendritic transport and receptor localization [16,17,18,19,20,21]. In addition, HTT seems to also be involved in autophagy, a major contributor to protein clearance and synaptic function [22,23,24,25,26]. Additional studies suggest a direct role of HTT in the regulation of transcription. Both wild-type and mtHTT proteins have been shown to directly associate with genomic DNA and also promote changes in chromatin states by their interaction with histone-modifying enzymes and key transcription factors ultimately leading to transcriptional dysregulation [27,28,29,30,31].

Recent reports have demonstrated that mtHTT levels and toxicity can be regulated by both posttranscriptional and posttranslational events. mtHTT-RNA undergoes an aberrant and incomplete splicing that generates HTT-exon1 transcripts and contributes to mtHTT toxicity [32,33]. Recently, it was shown that a class of small molecule splicing modifiers can promote selective splicing of an inducible HTT pseudoexon containing a premature termination codon (stop-codon psiExon), therefore reducing the production of HTT-exon1 mRNA and lowering mtHTT protein levels [34]. In addition, a large amount of evidence has shown that HTT posttranslational modifications (PTMs) have also the ability to regulate levels, localization, and toxicity of mtHTT. Although our knowledge about the enzymes responsible for the regulation of HTT PTMs is still under active investigation, they seem to provide a viable alternative as therapeutic agents. This review will focus on the role of kinases in regulating HTT phosphorylation, one of the most important PTM in regulating HTT toxicity, and their potential as therapeutic targets, with special emphasis on Protein kinase CK2, an emergent therapeutic target for the treatment of various neurodegenerative diseases including HD.

## 2. Current Therapeutic Approaches in HD

Currently, no disease modifying therapies are available for this devastating disease that results in death most commonly between 15 to 20 years after diagnosis [4]. The increasing number of reports highlighting the involvement of HTT in a large variety of different cellular and molecular processes as well as its interaction with other prone-aggregation proteins is a reflection of the complexity of the mechanisms involved in HD pathogenesis [28]. This complexity makes the identification of crucial pathways to be targeted for therapeutic purposes even more challenging. Though no curative interventions exist at this time,  there are a number of symptom-alleviating treatments currently in use and treatments currently in clinical trial stages. For depressive symptoms, antidepressants such as citalopram (Celexa), escitalopram (Lexapro), fluoxetine (Prozac), and sertraline (Zoloft) can be utilized [35]. Antipsychotic drugs such as quetiapine (Seroquel) and olanzapine (Zyprexa) and mood-stabilizing drugs such as divalproex (Depakote), carbamazepine (Tegretol), and lamotrigine (Lamictal) can be prescribed to quell violent outbursts and abrupt mood changes [35]. There are also drugs to control physical symptoms. For instance, tetrabenazine (Xenazine) and deutetrabenazine (Austedo) are reversible dopamine depleting agents with high affinity for vesicular monoamine transporter type 2 (VMAT2) and have been approved by the Food and Drug Administration (FDA) to treat chorea [35,36,37].

One of the most promising therapies against HD has been the development of antisense oligonucleotides (ASOs) against HTT. From a drug-discovery perspective, one major challenge regarding targeting HTT with small-molecule therapeutics has been due to the large size of HTT (>300 KDa) and that it lacks a characterized enzymatic function [38]. ASOs can bypass this issue by targeting a small region of the RNA to decrease its expression [38]. ASOs are generally between 12 and 30 nucleotides in length and can promote degradation of targeted RNA by several mechanisms depending on the designed binding structure [39]. Ionis Pharmaceuticals and its partner Roche initiated at the end of 2019 the very first clinical trial using ASOs (tominersen) to treat HD [40]. Though tominersen trials were recently halted in phase III due to negative effects, a post-hoc analysis of initial trials revealed a potential benefit specifically for young adults with HD [40,41]. Though a new clinical trial is still in development, its progress will undoubtedly be closely monitored by the HD research community. A summary of the current treatments and their molecular targets for HD is shown in Table 1.

Given that the current strategies for lowering HTT, while promising, have not yet achieved the desired success in clinical trials, a need for alternative therapies is imminent. With increasing evidence for the role of HTT PTMs in regulating mtHTT toxicity, abundance, and localization, therapies targeting the molecules that modify them have promise in becoming more successful therapeutic strategies.

## 3. Post-Translational Modifications and Their Role in HTT Toxicity

HTT PTMs have been shown to affect mtHTT toxicity and HD progression. PTMs do occur in wild-type HTT, but can become dysregulated in disease affecting aggregation, protein localization, mitochondrial function, protein interactions, axonal transport and other cellular functions that contribute to the toxicity of mtHTT and disease progression [65,66,67,68,69,70,71,72,73,74,75,76]. The molecules that regulate PTMs show promise as potential therapeutic targets as an alternative to aforementioned approaches that are focused on mtHTT lowering via ASOs or small molecules targeting HTT RNA splicing [34,38]. PTMs found on HTT include phosphorylation, acetylation, palmitoylation, SUMOylation, ubiquitination and methylation [67,69,70,72,77,78,79]. The study of HTT PTMs has been primarily focused on the first 17 amino acids on the N-terminal region of HTT (N17 region) and the poly proline rich domain (PRD) as these regions immediately precede and follow HTT exon 1, respectively, and whose structure have been shown to influence HTT exon 1 toxicity [80,81,82,83,84,85,86]. Recent studies using mass-spectrometry have identified several PTMs throughout the full length of the HTT protein, finding that specific PTMs cluster along the full-length protein in four distinct protease-sensitive regions or clusters [72,77]. These clustered PTMs and their main effects on HTT function and toxicity are summarized in Figure 1, and detailed below, along with several other important PTMs that do not fall within these clusters.

### 3.1. Phosphorylation

HTT phosphorylation is one of the most heavily studied PTM in HD, and thus there is an abundance of literature and information on identified phosphorylations. A summary of all identified HTT phosphorylations and their effects on HTT-mediated toxicity can be found in Table 2. Overall HTT phosphorylation tend to be decreased in the presence of expanded polyQ, which has pathological-associated effects on HTT nuclear localization, aggregation, and cellular toxicity [71,73,74,75,87,88,89,90,91,92,93]. Three highly studied phosphorylations on HTT are located within the N17 domain and correspond to T3, S13 and S16 [71,73,94]. Phosphorylation on these residues is overly decreased in HD. Hypophosphorylation of T3, S13 and/or S16 are associated with increased HTT aggregation, subcellular mislocalization, altered protein-protein interactions and enhanced toxicity [71,73,74,75,87].

**Table 2 biomedicines-10-01979-t002:** Summary of identified HTT phosphorylation sites, their effects and described kinases involved.

Residue	Effect	Implicated Kinase
T3	Effects mtHTT aggregation (conflicting reports enhance vs. prevention) [93,95] Reduces HTT mediated toxicity [93,95]	IKKβ [96]
S13	Decrease mtHTT toxicity [71,87,97,98,99] Decrease HTT aggregates [71,94,97,99] Promote HTT clearance [94,99] Increases nuclear localization [71]	TBK1 [99] CK2 [71,98] IKKβ [94,100]
S16	Decrease mtHTT toxicity [71,87,97,98] Decrease HTT aggregates [71,94,97] Promote HTT clearance [94] Increases nuclear localization [71]	CK2 [71,98] IKKβ [94]
S116	Decrease mtHTT mediated cell death in HEK293 cells and mouse primary cortical and striatal neurons [72,101]	Not described
S120	Reduce mtHTT levels in striatal and HEK293T cells [92] Influence neuropathology in HD mice [92]	NLK [92]
T271	Phospho-null mutation at this site decreases mitochondrial potential in HD HEK293 cells [72]	Not described
S421	Improves mitochondrial potential and form in hIPSCs [89]	AKT/SGK [102,103]
S431	Lack of phosphorylation on mtHTT increases cell viability and decreases mtHTT accumulation [104]	Not described
S432	Lack of phosphorylation on mtHTT decreases cell viability and increases mtHTT accumulation [104]	Not described
S434	Reduces caspase-mediated HTT cleavage [105]	CDK5 [105]
S536	Decreases HTT cleavage by Calpain and mtHTT toxicity [106]	Not described
S1181	Protects against polyQ induced p53-mediated cell death in response to DNA damage [107]	CDK5 [107]
S1201	Protects against polyQ induced p53-mediated cell death in response to DNA damage [107]	CDK5 [107]
S2076	Not described	ERK1 [106]
S2116	Impacts HTT structure and interaction with other proteins such as Polycomb Repressive complex 2 (PRC2) histone methyltransferase [108]	Not described
S2342	Modulation of mitochondrial potential [72]	Not described
S2489	Modulation of mitochondrial potential [72]	Not described
S2562	Reduces mtHTT cellular toxicity [72]	Not described
S2653/2657	Not described	ERK1 [106]

Mass spectrometry analyses in cell models transfected with HTT exon 1 fragments have shown that T3 phosphorylation (pT3) is one of the most prevalent phosphorylations present in exon 1 [93]. Interestingly, there is an inverse correlation between the CAG repeat length and the levels of pT3 [93]. Studies in striatal progenitor cells (ST14A) transfected with HTT exon 1 containing different CAG repeat length showed that HTT-pT3 decreased in those cells expressing longer CAG-repeat tracks. Similarly, decreased levels of HTT-pT3 are also found in the striatum of CAG140 mice, although an enhanced HTT-pT3 ratio was found in the cortex. On the other hand, ST14A cells and drosophila models of HD expressing a phosphonull (T3A) or a phosphomimetic (T3D) HTT-exon 1 showed that T3D enhanced HTT aggregation. Unexpectedly, despite HTT-exon1-T3D providing enhanced aggregation, it also ameliorated neurodegeneration and extended the lifespan of flies compared to HTT-exon1 [94]. In a more recent study using fluorescent bimolecular complementation in drosophila models, T3D mutant prevented HTT aggregation but not its oligomerization strengthening the idea that T3 hypophosphorylation plays a critical role in modulating HTT toxicity [95].

Phosphorylations on S13/S16 have also been shown to improve HD-like phenotypes in mouse models that have a HTT-S13/16 phosphomimetic mutation [97]. However, increasing evidence shows that the phospho mimicking mutations do not reproduce all aspects of protein phosphorylation, and recent studies have shifted to using methods that allow the study of *bona fide* phosphorylations by introducing site-specific phosphorylations on the N17 HTT region. In this context, phosphorylation at S13 or S16, but not pseudo phosphorylation, results in a significant retardation of mtHTT exon 1 aggregation [109]. Additional studies using *bona fide* phosphorylations have revealed beneficial effects of site-specific induced phosphorylation including decreased mtHTT aggregation in response to T3, S13, and S16 phosphorylation [74,98,110].

Phosphorylations outside of the N17 region also influence HTT abundance, aggregation, and toxicity properties [88,89,90,91,92,105,107]. For instance, phosphorylation at S120 in mtHTT is decreased in HD and it is associated with mtHTT ubiquitination and clearance [92]. This is supported by studies in which expression of a phosphonull mutant (S120A) in HEK293T cells leads to enhanced mtHTT accumulation [92]. Furthermore, S421 phosphorylation, which is also decreased in HD models [88], has been shown to be protective in HD human induced pluripotent stem cells (hIPSC), influencing mitochondrial function and phenotype [89]. S421 phosphorylation also restores the axonal transport function of HTT that is impaired in mtHTT expressing cells [90] and promotes mtHTT clearance in BACHD mice [91]. Phosphorylation at S434 is also decreased in HD transgenic mice expressing the first 171 residues of HTT [105]. S434 phosphorylation was shown to decrease mtHTT cleavage and thus decrease aggregation and toxicity in SK-N-SH cells transfected with HTT fragments and PC-12 cells expressing full length HTT [105]. Lastly, phosphorylations at S1181 and S1201 have been shown to protect against mtHTT induced toxicity and the absence of these phosphorylations in wtHTT induces toxicity [107].

All this evidence demonstrates that increasing the phosphorylation of HTT could be a potential venue to ameliorate mtHTT toxicity and improve HD-like phenotypes. This therapeutic potential could be achieved by strategies targeting phosphatases to prevent HTT hypophosphorylation or by targeting key kinases to modulate HTT phosphorylation. Since knowledge on HTT-specific phosphatases is very limited [111,112], kinases that are known to influence site-specific phosphorylations are more promising therapeutic targets and are the focus of this review.

### 3.2. Other PTMs

HTT is subjected to several other PTMs besides phosphorylation and a brief summary of other PTMs is described below:

#### 3.2.1. Acetylation

Acetylations of multiple lysine residues in the HTT protein have been identified, including lysines 6, 9, and 15 in the N17 region [76] and lysine 444 [72,79]. While there is conflicting information on whether specific acetylations on HTT are beneficial or detrimental to HD, acetylation has been shown to impact mtHTT trafficking, clearance, aggregation, and interactions with lipidic membranes [65,72,76,79]. Molecules identified to mediate acetylation in HD include N-terminal actyltransferase (NAT) [65], Creb binding protein (CBP) [79] and Histone deacetylase 1 (HDAC1) [79,113] and may all be potential therapeutic targets to modulate HTT acetylation.

#### 3.2.2. Palmitoylation

Palmitoylation is the covalent attachment of fatty acids, such as palmitic acid, to Cys and less efficiently to Ser and Thr, catalyzed by a family of palmitoyl acyltransferases (PATs). HTT is palmitoylated at Cys214 by the huntingtin-interacting protein 14 (HIP14) and 14-like (HIP14L) acyltransferases [20,67,114]. HTT palmitoylation has been shown to be reduced in various HD mouse models including YAC128, BACHD, Hu97/18 and Hu128/21, and in lymphoblastoid cells derived from HD patients, and the reduction in palmitoylation correlate with the CAG- length [66,67]. Inhibition of acyl-proteinthioesterase leads to increase palmitoylation associated with decrease mtHTT toxicity and aggregation in primary neurons from YAC128 mice, in COS-7 cells expressing HTT-Q128, in HD patient derive lymphoblasts, and in CAG140 mice [66,114]. These results provide evidence that restoring HTT palmitoylation in HTT via activation of HIP14 and/or HIP14-like protein (HIP14L) or by inhibition of APT could lead to a successful therapeutic strategy to decrease mtHTT toxicity.

#### 3.2.3. SUMOylation

SUMOylation consists of the addition of small ubiquitin-like modifier (SUMO) polypeptides to internal Lys residues of target proteins. HTT SUMOylation has been reported to by dysregulated in HD [78] and seems to be largely mediated by the Protein Inhibitor of Activated STAT (PIAS) [115] and Ras homolog enriched in striatum (Rhes) [116]. SUMOylation of HTT affects the solubility and toxicity of HTT [78,116,117,118,119], but conflicting reports on the effect of this PTM in HD warrants further investigation [68].

#### 3.2.4. Methylation

Methylation consists of the addition of methyl groups to the nitrogen-containing sidechains of Arg and Lys and it is mediated by methyltransferases. Arginine methylations are mediated via arginine methyltransferase (PRTMs) 4 and 6 (PRTM4 and PRTM6) which have been identified as mediators of identified methylations of HTT [69,70]. Methylation of HTT affects HTT toxicity and solubility and impair axonal trafficking [69,70]. Very little is known about methylation sites in HTT, but their identification has garnered increased attraction.

## 4. Kinases Involved in HTT Phosphorylation and Their Potential as Therapeutic Agents

Understanding the specific kinases that regulate HTT phosphorylation could result in the identification of effective therapeutic targets to treat HD. Kinases that have been implicated in HTT phosphorylation include, but are not limited to: Tank-binding kinase 1 (TBK1), Inhibitor of nuclear factor kappa-B kinase subunit beta (IKKβ), Cyclin-dependent kinase 5 (CDK5), Nemo-like kinase (NLK), and Protein kinase 2 (CK2, formerly known as Casein kinase 2) [71,94,96,99,100,105,107]. While these kinases are not the only kinases linked to HD and/or HTT phosphorylation, they were chosen as a representative example to highlight the importance of studying kinases as potential therapeutic agents in HD. However, while these kinases are linked to the phosphorylation of specific residues on HTT, the effects of targeting these kinases in the overall HD pathophysiology need careful interpretation since these kinases can phosphorylate a myriad of other targets. Figure 2 summarizes identified phosphorylations sites and the kinases implicated, along with putative phosphorylation sites based on kinase consensus sequences. A description of key kinases involved in HTT phosphorylation are detailed below:

### 4.1. IKKβ

IKKβ belongs to the IKK family of kinases, and it has been implicated in mediating HTT phosphorylation at S13, S16, and T3 [94,96,100]. IKKβ-mediated HTT phosphorylation has been implicated in increased HTT clearance in cell and mouse models of HD [94,100]. Conditional tamoxifen-inducible IKKβ knockout mice crossed with R6/1 mice significantly worsened HD pathological phenotypes by altering autophagy and enhancing striatal degeneration. The increased behavioral pathology in R6/1 mice lacking IKKβ directly correlated with reduced levels of HTT-pS13, indicating that IKKβ is involved in mediating HTT-pS13 and that such phosphorylation is necessary to prevent mtHTT toxicity [100]. Contrarily, pharmacological or genetic inhibition of IKKβ using siRNAs in murine immortalized ST*Hdh* cells resulted in more phospho-N17 HTT and enhanced HTT nuclear accumulation and aggregation but did not affect cellular toxicity [71]. Further analyses are warranted to determine the therapeutic potential of targeting IKKβ.

### 4.2. TBK1

TBK1 also belongs to the IKK-kinase family and was identified in a kinase-based screening assay for its role in phosphorylating HTT, specifically at S13 [99]. TBK1-mediated HTT-pS13 was shown to influence subcellular localization of mtHTT and decrease aggregation and cytotoxicity in primary neurons and *C. elegans* models of HD [99]. When IKKβ-driven phosphorylation was compared to TBK1, TBK1 was found to be more efficient in phosphorylating HTT in vitro. However, when TBK1 was knocked down in primary striatal cultures, phosphorylation still occurred, likely by the action of IKKβ or other implicated kinases [99]. Nevertheless, the study by Hegde et al., suggested that upregulation and/or activation of TBK1 may represent a viable strategy for the treatment of HD by simultaneously lowering mutant HTT levels and blocking its aggregation.

### 4.3. CDK5

CDK5 is a proline-directed serine/threonine kinase that belongs to the family of cyclin-dependent kinases. CDK5-mediates phosphorylation of HTT at S434, S1181, and S1201 which prevent aggregation and toxicity of mtHTT [105,107]. CDK5 was found elevated in HEK293 cells expressing polyQ-HTT which can be interpreted as a protective attempt to cope with mtHTT toxicity [129]. However, a recent study in HdhQ7/Q111 mice lacking one allele of *Cdk5* displayed improved cognitive function and motor behavior [130], which is at odds with the protective effect previously associated to CDK5 [105,107]. While Alvarez et al. did not test the impact of *Cdk5* haploinsufficiency in HTT phosphorylation, the study suggests that other CDK5 targets, such as Src kinase, which impacts the localization of the NMDA receptor subunit NR2B, may play a more important role in disease progression than HTT phosphorylation.

### 4.4. NLK

NLK is a proline-directed serine/threonine MAP kinase [131]. A large mass spectrometry screening in tissue from bacterial artificial chromosome (BAC)-mediated transgenic HD mice (BACHD) first identified NLK as interacting with mtHTT [132]. NLK and mtHTT interaction has been further confirmed via co-immunoprecipitations in HEK293-FT cells co-transfected with Flag-NLK and full-length HTT containing Q123 or Q23 and in zQ175 HD mice [92]. NLK is involved in phosphorylating HTT at S120, a protective PTM [92]. NLK is decreased in postmortem cortex samples from HD patients, in the striatum of N171-82Q and HdhQ250 HD mice, and also in ST*Hdh* Q111 cells which is thought to contribute to mtHTT toxicity [92]. Overexpression of NLK via adeno-associated virus (AAV) injection into the striatum of N171-82Q mice leads to decreased brain atrophy, increased dopamine- and cAMP-regulated neuronal phosphoprotein (DARPP32) expression and decreased mtHTT aggregates. On the contrary, NLK haploinsufficiency in HdhQ250 mice showed increased brain atrophy and mtHTT levels as well as decreased DARPP32 [92]. Given that the HEK293-FT cells transfected with NLK had higher levels of ubiquitination and that inhibition of the proteasome with epoxomicin abolished NLK-mediated mtHTT lowering, it was suggested that NLK mediates mtHTT depletion by enhancing its ubiquitination and degradation [92]. While thus far only one study has examined the effects of NLK in HD, this study provides ample evidence of a positive effect of increasing NLK in HD models. Therefore, NLK appears to hold promise as a potential therapeutic target.

### 4.5. CK2

CK2 is a serine/threonine protein kinase extensively studied in the context of cancer and inflammation [125,133] and it was first identified in relation to HD in studies using HEK293T cells co-expressing NMDA receptors (NR1 and NR2B) and human HTT-138Q where CK2 was seen to be upregulated [129]. This observation has since been validated in multiple HD models including striatal tissues from YAC72, YAC128, and zQ175 mice, in primary neurons from YAC mice, in immortalized ST*Hdh* Q111 cells, and in striatal tissues from patients with HD [129,134,135]. The study by Fan et al. showed that pharmacological inhibition of CK2 exacerbated NMDA-mediated toxicity in YAC-derived neurons and proposed a protective role of CK2 in HD. It was later suggested that CK2 plays a role in phosphorylating HTT at S13/16 and its pharmacological inhibition decreased nuclear pS13/16 and increased toxicity [71]. More recently, N6-Furfuryladenine (N6FFA), a product of oxidative DNA damage, has been shown to be used by CK2 to promote N17 phosphorylation, when the region is previously primed by pS13. N6FFA is salvaged by adenine phosphoribosyltransferase (APRT) to a triphosphate and is then used as a phosphate donor by CK2. Interestingly, N6FFA treatment of YAC128 HD mice showed increased soluble HTT in cortical regions and improved motor behavior [98].

It is important to mention that the studies where the protective role of CK2 in HD was investigated were conducted almost exclusively using treatment by benzimidazole derivatives (DRB and DMAT), which not only inhibit CK2 but also potently inhibit several additional kinases, RNA Pol II transcription, and DNA replication [136,137,138,139]. Therefore, the specificity for CK2 inhibition in these studies is unclear. In addition, while several CK2 consensus sequences have been identified along HTT (Figure 2), the N-terminal sequence of HTT containing S13/16 (KAFE**S_13_**LK**S_16_**FQQQ) lacks a CK2 consensus sequence (SxxE/D) [140,141]. Although CK2 can phosphorylate substrates without this minimum consensus site [125], genetic evidence for the role of CK2 in the direct phosphorylation of HTT is still lacking, hence the protective role of CK2 in HD via HTT phosphorylation is still questionable. CK2 is composed of four subunits: two beta regulatory subunits (CK2β) and varying combinations of two catalytic subunits, alpha (CK2α) and/or alpha-prime (CK2α’) [125,133]. Recent studies in ST*Hdh*-Q111 cells using siRNA against different CK2 subunits showed that silencing the catalytic subunit CK2α’, but not CK2α or CK2β, improved cellular toxicity [134]. Similarly, CK2α’ genetic haploinsufficiency in zQ175 mice (zQ175:CK2α’^(+/−)^) provided ample benefits in ameliorating several HD-like phenotypes suggesting that CK2α’ plays a detrimental role in HD [134,135]. Unfortunately, these studies did not explore whether the benefits exerted by CK2α’ in these various HD models had an impact on HTT phosphorylation. Although inhibition of CK2 has been proposed as a potential therapeutic strategy in other neurodegenerative diseases [125,142], its role in HD has remained controversial.

## 5. Other Roles of Kinase CK2 in HD

As previously discussed, CK2 holoenzyme is composed of two different catalytic subunits, CK2α and CK2α’ [125,133]. In contrast to CK2α, which is highly expressed throughout the body and is an essential protein with hundreds of targets, CK2α’ has very few identified substrates [140,143,144] and its expression is more restricted to the testes and brain [145]. These differences imply that both catalytic subunits might have fundamentally different roles in the regulation of cellular processes. Despite these differences, both subunits present a high structural homology and share ~100% homology in their ATP binding site [133]. The structural similarities between CK2α and CK2α’ have made it impossible to distinguish their differential contribution using pharmacological approaches in different contexts, including HD. Pharmacological inhibitors against CK2, are often based on benzimidazole derivatives that act as ATP analogs and they simultaneously inhibit both catalytic subunits [139,146,147] (further discussed below). A differential abundance between CK2 subunits in HD was first reported in YAC72 and YAC128 striatal tissues although their differential contribution to HD remained unknown [129]. Almost a decade later Gomez-Pastor et al. showed that CK2α’, but not CK2α or CK2β, was specifically induced in HD cells and mouse models and in patients with HD, and that such induction played a detrimental role in HD [134,135]. Although, whether CK2α’ plays a direct role in HTT phosphorylation still remains elusive, other roles for this kinase have been recently discovered in the context of HD. The various identified substrates for CK2α’ and the effects of CK2α’ mediated phosphorylation in the context of HD are summarized in Figure 3.

### 5.1. CK2α’ Facilitates Protein Homeostasis in HD

Previous studies reported that mtHTT impairs the ability of cells to activate the Heat Shock Response (HSR), a molecular pathway governed by Heat Shock Factor 1 (HSF1) that is essential to induce molecular chaperones (Heat Shock Proteins, HSPs) that facilitate protein homeostasis and prevent protein aggregation (reviewed by Gomez-Pastor et al., 2018 [148]). Several HSPs are downregulated in cells and mouse models of HD, which is known to contribute to mtHTT aggregation [134,149,150]. HSF1 depletion has also been reported in several cell and mouse models of HD, in human iPSCs derived from patients with HD differentiated into MSN-like cells, and in human postmortem striatum from patients with HD [134,151,152,153,154]. Initial studies proposed that down-regulation of HSPs in HD was facilitated by mtHTT-mediated changes in chromatin remodeling [149]. More recently, it was demonstrated that an abnormal mtHTT-mediated proteasomal degradation of HSF1 also contributes to the impaired ability of HD cells to trigger the HSR in response to HTT aggregation [134].

Upregulation of CK2α’ in HD is involved in the pathological degradation of HSF1 [134]. CK2α’ directly mediates the phosphorylation of HSF1 at S303 and S307, two PTMs previously associated with the inactivation of HSF1 [155,156,157,158]. These phosphorylations inactivate HSF1 and allow the recruitment of the E3 ligase Fbxw7, also upregulated in HD, which ubiquitinates HSF1 and signals the protein for proteasomal degradation [134]. HSF1 degradation leads to decreased expression of several HSF1 targets, including chaperones such as Hsp70 and Hsp25, genes related with GTPase function, and several synaptic proteins [134,153,159]. Genetic depletion of CK2α’ via siRNAs in HD cells or gene haploinsufficiency in zQ175 mice (CK2α’^(+/−)^) decreased S303 and S307 phosphorylation, increased HSF1 protein levels and HSPs expression, and subsequently decreased mtHTT aggregation [134]. CK2α’ haploinsufficiency in zQ175 mice also resulted in improved synaptic function, increased MSN spine maturation, and ameliorated motor deficits [134,135], thus supporting a role for CK2 α’ selective inhibition as a potential therapeutic option for HD.

### 5.2. CK2α’ Associates with Increased Striatal Inflammation in HD

CK2 regulates signaling pathways involved in inflammation [160,161,162]. Inhibition of CK2 has shown to deplete pro-inflammatory cytokines such as IL-6 [163]. IL-6 and other cytokines are increased in HD and are often used as a marker of disease progression [164]. While the CK2α subunit is most commonly associated as the main driver of CK2-mediated inflammation [160], a role for Ck2α’ has recently been highlighted in ST*Hdh*-Q111 cells treated with siCK2α’ and in zQ175 mice with CK2α’ haploinsufficiency [135]. Knock-down of CK2α’ in Q111 cells resulted in decreased IL-6 mRNA levels. In addition, zQ175:CK2α’^(+/−)^ mice showed decreased levels of proinflammatory cytokines compared to zQ175 mice including TNF-α, IL-27, IL-7 and others, as well as decreased astrocyte pathology in the striatum [135]. This study highlights a role for CK2α’ in mediating some of the inflammatory processes seen in HD.

### 5.3. CK2α’ Contributes to the Dysregulation of Expression and Function of Synaptic Genes in HD

Excitotoxic damage from dysregulation of glutamatergic signaling is hypothesized to contribute to HD pathogenesis [165]. RNA-seq studies in zQ175 mice lacking one allele of CK2α’ showed an overall restoration of transcriptional deficits associated with HD and revealed a role of CK2α’ in synaptogenesis and in the regulation of several genes involved in glutamatergic signaling [135]. Among the most differentially dysregulated genes by CK2α’ are *Slc30a3* (Solute Carrier Family 30 Member 3, a zinc transporter (ZnT3) selective for glutamate synapses), *Grm2* (Glutamate Metabotropic Receptor 2), *Slc17a7* (Solute Carrier Family 17 Member 7; alias VGlut1), and *Nrp2* (Neuropilin 2). Restored expression of these genes in zQ175:CK2α’^(+/−)^ mice also correlated with an improved frequency of striatal mEPSCs (miniature excitatory postsynaptic currents) observed by reducing CK2α’ levels [135]. Since direct regulation of these genes by CK2α’ is unlikely, other factors involved in transcriptional regulation might be directly regulated by the action of CK2α’ (discussed below).

In addition to the transcriptional regulation of glutamate-signaling related genes, CK2 also controls the function and localization of other genes involved in glutamatergic signaling such as NMDARs. CK2 directly phosphorylates S1480 of the NR2B subunit and such phosphorylation limits surface NMDAR expression by disrupting interactions between the receptor and the postsynaptic scaffolding protein PSD-95 [166]. In addition, PSD-95 itself is a CK2 substrate [167]. Extensive studies have addressed the role of CK2 in the regulation of NMDARs in various contexts [167,168,169,170,171], although our knowledge of such regulation in HD and whether there is a differential contribution between CK2α and CK2α’, remains limited. Pharmacological inhibition of CK2 in primary neurons from YAC128 mice co-expressing NMDAR and human Htt-Q138 exacerbated NMDAR-mediated excitotoxicity and suggested a protective role of CK2 in preventing excitotoxic damage [129]. However, this hypothesis is at odds with more recent literature suggesting that CK2, and more specifically CK2α’, play a detrimental role in HD [134,135]. It has been reported that CK2α’ haploinsufficiency in zQ175 mice restored PSD-95 levels in the striatum and increased the frequency of miniature excitatory postsynaptic currents (mEPSCs) [135]. Therefore, an assessment of potential alterations in NMDAR abundance and distribution in zQ175:CK2α’^(+/−)^ mice are warranted to shed light on whether this particular subunit is involved in NMDAR regulation.

### 5.4. CK2α’ Promotes Synucleinopathy in HD

Alongside the toxicity mediated by mtHTT protein and its aggregates is its pathogenic interactions with other aggregation-prone proteins such as alpha-Synuclein (αSyn) [135,172,173,174,175], a protein more commonly associated with Parkinson’s disease (PD) and Lewy body disease (LBD) [176,177]. αSyn is a cytoplasmic protein typically associated with synaptic function and vesicle budding, although the exact function is unclear [177]. αSyn interacts with HTT and it has been shown to colocalize with HTT aggregates in several HD mouse models, and in the striatum and cortex of HD patients [135,172,173,174]. αSyn is also found at increased levels in HD patient serum and post-mortem putamen samples, suggesting that αSyn dysregulation could contribute to HD pathogenesis [175,178]. This is supported by studies in R6/1 mice lacking αSyn showing improved autophagy and amelioration of motor deficits [173,179]. Although the mechanism by which αSyn contributes to HD remains elusive, recent studies indicate that αSyn participates in the transcriptional dysregulation of glutamate-receptor related genes in a mechanism connected to CK2α’ [135].

CK2α’ promotes αSyn phosphorylation at S129 (pS129-αSyn) in HD [135]. This PTM is associated with αSyn toxicity, nuclear mislocalization, transcriptional dysregulation and synucleinopathy in PD and other neurological disorders where the role of CK2 has also been proposed [135,180,181,182,183,184,185,186,187]. We showed that pS129-αSyn levels are increased in the striatum of patients with HD, as well as in zQ175 mice [135]. In zQ175;CK2α’^(+/−)^ mice, pS129-αSyn levels are decreased compared to zQ175 mice, and this is associated with the restoration of glutamate-related synaptic genes dysregulated in zQ175 mice, as well as improved motor behavior and synaptic function [135]. All this evidence suggests that CK2α’ could be a potential therapeutic target in the amelioration of HD and perhaps other neurodegenerative diseases.

## 6. CK2 in Other Neurodegenerative Disorders

CK2 has been implicated in a range of neurodegenerative disorders including Alzheimer’s disease (AD), Parkinson’s disease (PD), Amyotrophic lateral sclerosis (ALS) and Spinocerebellar ataxia type 3 (SCA3) and has emerged as a potential therapeutic target across multiple NDs [125] (Figure 4). Since this topic has been recently reviewed by Borgo et al., 2021 [125], we will only briefly discuss it here.

### 6.1. Alzheimer’s Disease and Other Tauopathies

Increased levels of CK2 have been detected in affected brain regions in patients with AD [163] and was found to immunocolocalize to the paired helical filaments (PHFs) of the neurofibrillary tangle (NFT)-bearing neurons, as well as to PHF in neuropil threads and some dystrophic neurites in plaques [188]. Increased CK2 has been associated with the phosphorylation and alteration of very different targets in AD. For instance, CK2 has been implicated in mediating tau phosphorylation, one of the primary hallmarks of all tauopathies, which contributes to the formation of neurofibrillary tangles [189].

A kinase screening in a N2a cell model of tauopathy looking at potential kinases involved in tau aggregation revealed CK2 as one of the kinases with the most profound effects in modulating tau aggregation [189]. The role of CK2 in tau phosphorylation resides in its ability to phosphorylate and subsequently activate I2PP2A/SET, a potent inhibitor of phosphatase PP2A, resulting in tau hyperphosphorylation [190]. Interestingly, tau pathology has also been observed in HD. For instance, increased tau protein levels were observed in cortex and CSF of HD patients [191], as well as increased tau phosphorylation [178,192] correlating with disease burden [164]. An increase in soluble truncated tau species (DTau314) related to cognitive dysfunction in AD has also been shown in the striatum and cortex of patients with HD [193]. However, whether CK2 (either α or α’) is meaningfully involved in tau pathology in HD remains unknown.

Contrary to what was proposed in HD, CK2 contributes to NR2B phosphorylation and to a detrimental imbalance in NR2B synaptic/extrasynaptic ratio in AD, which is also associated with tau accumulation [170]. In addition, CK2 participates in the secretion of pro-inflammatory cytokines, assessed in human astrocytes, and was suggested to modulate neuroinflammation in AD, although the mechanism by which CK2 participates in such process is unclear [163]. Interestingly, C57/BL6 mice injected with AAVs overexpressing CK2 into the hippocampus show cognitive impairments, similar to those observed in AD mice, along with increased tau pathology [190]. Although all this evidence suggests that inhibition of CK2 might lead to beneficial effects in AD, other studies have suggested the opposite due to a role of CK2 in the phosphorylation of 5-HT4 receptors and 5-HT4-mediated reduction in amyloid-beta production [125,194,195].

### 6.2. Parkinson’s Disease and Synucleinopathy

As discussed above, phosphorylation of αSyn at S129 (pS129-αSyn) is a hallmark of PD and other synucleinopathies [177,180]. Several kinases have been proposed to mediate pS129-αSyn in PD, including CK2 [125,142,187,196]. Commonly used lipid-lowering drugs such as Lovastatin, have shown an impact in decreasing motor symptom progression in patients with PD and they are beneficial for reducing the risk of PD [197,198,199]. A recent study showed that the underlying mechanism by which lovastatin mediates protective effects in PD is by inhibiting CK2 activity and decreasing pS129-αSyn [200].

CK2, specifically CK2α, has also been associated with L-DOPA-induced dyskinesia (LID) in PD. L-DOPA is one of the most effective pharmacological treatments in PD but its long-term use generates a loss in efficacy and LID [201]. Conditional knockout mice lacking CK2α in striatonigral neurons showed a reduction of dyskinesia, but when knocked out in striatopallidal neurons LID was increased [201]. These results add to the list of the negative effects mediated by CK2 in PD. Further investigation will be necessary to assess whether CK2α’ also plays a role in the pathophysiology of PD and other synucleinopathies.

### 6.3. Amyotrophic Lateral Sclerosis

CK2 phosphorylates Trans-activation Response DNA-binding Protein-43 (TDP-43), a protein associated with ALS and other NDs [202,203]. In vitro studies showed that CK2-dependent phosphorylation of TFP-43 promotes TDP-43 polymerization into structures that resemble those from FTLD-TDP (Frontotemporal lobar degeneration with TDP-43 inclusions) [203]. However, recent studies have shown that other kinases are more efficient in phosphorylating TDP-43 in the context of ALS (revied in Eck et al., 2021 [204]).

### 6.4. Spinocerebellar Ataxia

Ataxin-3, the protein responsible for spinocerebellar ataxia 3 (SCA3), has multiple phosphorylation sites and specific phosphorylation patterns can decrease or increase its aggregation. Ataxin-3 phosphorylation mediated by CK2 increases protein accumulation in the nucleus, and inhibition of CK2 by DMAT or TBB in SCA3 cell models results in lower nuclear inclusions [205,206]. This topic has been recently reviewed by both Chen et al., 2020 [207] and Borgo et al., 2021 [125].

Based on the evidence provided above, it is clear that upregulation of CK2 and the mediated effects could represent a convergent mechanism of neurodegeneration across multiple NDs and therefore targeting CK2 has therapeutic potential for the treatment of several NDs. It also highlights that different CK2 catalytic subunits may play different roles in the pathogenesis of those NDs and therefore having the ability to selectively inhibit CK2α and CK2α’ is critical for the development of effective therapeutic strategies to target CK2 in neurodegeneration, and more specifically in HD.

## 7. Current CK2 Inhibitory Strategies and Their Potential as Therapeutic Agents in Neurodegeneration

In recent years, more potent and specific CK2 inhibitors to those derived from carboxyl acids, and discussed above, have been developed. Figure 5 summarizes some common CK2 inhibitors and some of their off target effects including; (i) 11-mer peptide (PC) [208] (ii) polyoxometalates (POMs) aggregates of early-transition metal ions and oxo ligands [209], (iii) CIGB-300 a peptide based inhibitor [210], (iv) CX-4945 a small molecule ATP-competitive inhibitor [210], (v) 2-dimethylamino-4,5,6,7-tetrabromo-1H-benzimidazole (DMAT) an ATP Competitive inhibitor is a Benzomidazole derivative [210] (vi) Emodin is a Anthraquinone derivative naturally occurring in plants such as the Rheum palmate [210,211], (vii) 4,5,6,7-tetrabromobenzotriazole (TBB) an ATP Competitive inhibitor is a Benzomidazole derivative [210]. Pyridoquinoline CX-4945 (Silmitasertib) is the first CK2 inhibitor (IC_50_ 1 nM) to be safely administered in vivo and currently is in human clinical trials for various forms of cancer [212,213,214]. This compound showed high oral bioavailability (20%), no detectable mutagenicity or genotoxicity, and the ability to cross the blood-brain-barrier (BBB) [214,215]. Although some off-target effects have also been described for CX-4945 [216,217] they are very limited, making it a good candidate for CK2 inhibitory studies in vivo. Another recent CK2 inhibitor is a peptide-based molecule CIGB-300, which is a cyclic cell permeable peptide also in clinical trials to treat cancer [125,147,210]. Similar to the context of neurodegeneration, there is a debate in the cancer field on the potential differential contribution of the different CK2 subunits to oncogenesis where CK2α seems to play a more relevant function than CK2α’ [125]. Although the CK2 inhibitors currently available for human studies provide an immediate opportunity for their use as potential pharmacological agents in neurodegeneration studies, there is still the caveat of lack of selectivity targeting the different CK2 catalytic subunits. Therefore, further studies are needed to implement drug discovery strategies that can lead to selective inhibition of CK2α’ vs. CK2α for their potential use as therapeutic agents in the treatment of HD and perhaps other NDs.

## 8. Conclusions

CK2 inhibition holds promise as a potential therapy, not only for HD, but for other neurodegenerative disorders. However, there are still gaps in knowledge and conflicting reports that need resolution to better inform the effectiveness of CK2 inhibition in HD. These include: (1) use of more genetic approaches to exam the effects of selectively targeting CK2α or CK2α’ in vivo, (2) evaluation of the impact of CK2 depletion in several HD mouse models, and (3) development and optimization of selective CK2α’ and CK2α inhibitors to test their efficacy in modulating mtHTT toxicity. As discussed in this review, there is a large amount of evidence indicating that CK2α’ mediates detrimental effects in HD at the level of protein homeostasis, inflammation, synaptic gene expression, synucleinopathy and motor behavior. Although pharmacological inhibition of CK2 has provided detrimental effects in HTT phosphorylation, the current research has not resolved whether there is a differential contribution of the different CK2 catalytic subunits to this effect. We therefore suggest CK2α’ specific inhibition as a potential therapeutic strategy that could provide long-lasting beneficial effects in ameliorating several HD-related phenotypes.

## Figures and Tables

**Figure 1 biomedicines-10-01979-f001:**
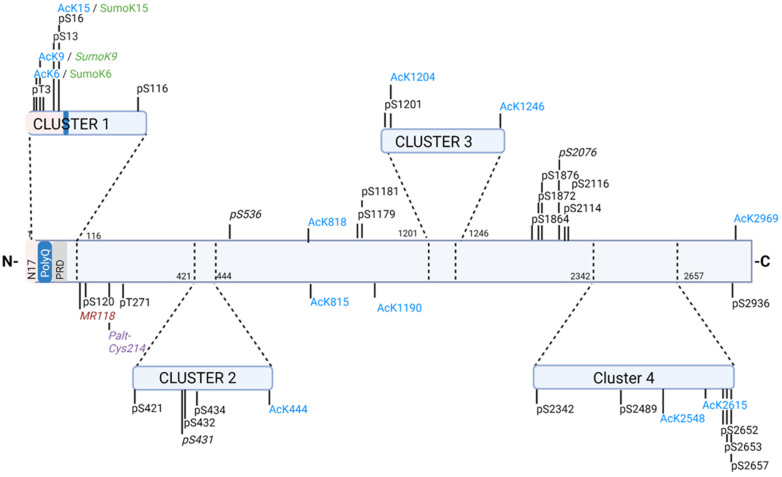
Post-translational modifications of HTT. Arbez et al. 2017 [72], identified clusters of phosphorylations and acetylations that clustered around protease sensitive domains. These clusters were defined by a scoring system defined in Arbez et al. 2017 for the PTMs examined based on functional effects of mtHTT toxicity. In addition to those identified phosphorylations and acetylations seen to cluster in Arbez et al. 2017, other identified PTMs that have been shown to have an effect on HD/HTT toxicity in some capacities are included, more details on the effects for phosphorylations can be found in Table 2 (those italicized indicate PTMs not originally analyzed to belong to the clusters by Arbez et al. 2017). PTMs within the clusters can have summarized/generalized effects as follows (direction of these effects, whether beneficial or detrimental depends on the specific PTM); *Cluster 1:* Includes the N17 and PolyQ regions, along with exon 1. PTMs here largely affect nuclear localization, mtHTT aggregation, and mtHTT toxicity. *Cluster 2*: Located between N terminal domains I and II. PTMs here largely show effects on mtHTT aggregation and toxicity, mitochondrial phenotype, and caspase cleavage. *Cluster 3:* Located between N-terminal domain II and C-terminal domain I. The one phosphorylation in this cluster, S1201 has been shown to protect against polyQ induced p53 mediated cell death, the acetylations may modulate mtHTT toxicity and mitochondrial potential changes. *Cluster 4:* Located in C-terminus domain II. PTMs here effect mitochondrial potential and can modulate mtHTT toxicity [72]. (Abbreviations and color code p (Black) = phosphorylation, Ac (Blue) = Acetylation, Palt (Purple) = palmitoylation, M (Red) = Methylation). Created with Biorender.com (accessed: 29 June 2022).

**Figure 2 biomedicines-10-01979-f002:**
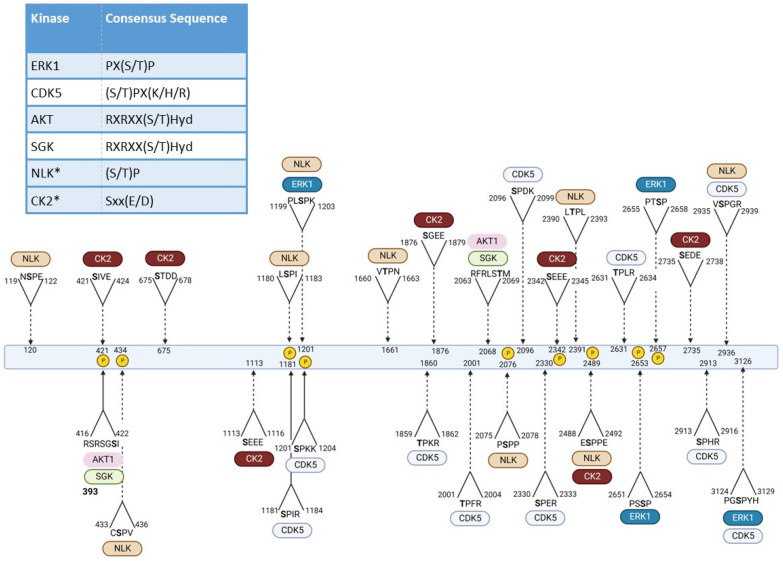
Identified kinase consensus sites along HTT. Consensus sites for ERK1, CDK5, AKT, SGK, NLK and CK2 are shown in top left panel [120,121,122,123,124,125]. IKK or TBK1 were not included due to the lack of well-defined consensus phosphorylation sequences for these kinases. Putative phosphorylation sites were identified for several kinases implicated in HTT phosphorylation along HTT and confirmed via https://www.phosphosite.org/ (accessed 9 August 2022) [126]. * CK2 and NLK returned 25+ putative sites and only those nearby previously identified phosphorylation sites or in critical protein regions are shown. For CK2 only sites that did not include R, K or P at the +1 position and at least 2 D or E’s in the sequence were included [127]. For NLK, only those sites that better aligned with the phosphosite Sequence logo were included. Putative kinase sites were then identified on HTT using https://www.bioinformatics.org/sms2/protein_pattern.html (accessed 9 August 2022) [128], using human HTT as reference (NCBI NP_002102.4). Putative sites for which phosphorylations have been identified are marked with a yellow P and details for these sites can be found in Table 2. Dashed lines indicate a kinase identified via this consensus sequence, but no evidence connecting the kinase to the site yet. Solid line indicates that the consensus sites were identified in regions where phosphorylation has been confirmed, with the kinase implicated in mediating the phosphosphorylation. Created with Biorender.com (accessed 10 August 2022).

**Figure 3 biomedicines-10-01979-f003:**
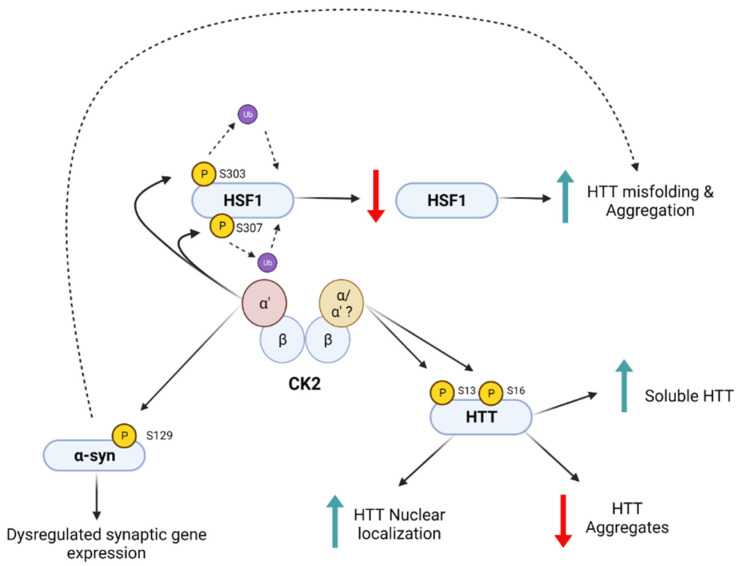
CK2 Targets in Huntington’s Disease. CK2 mediates the phosphorylation of several targets identified to be relevant to HD: HTT, HSF1, and a-syn. Gomez-Pastor et al., 2017 and Yu et al., 2022 [134,135] have identified the a’ subunit to be the primary catalytic subunit driving HSF1 and α-syn phosphorylation, respectively, although while evidence from Atwal et al., 2011 and Bowie et al., 2018 [71,98] support CK2 phosphorylating HTT the respective contributions of the alpha and alpha prime subunits, if any difference, have not been deciphered. Downstream effects, as summarized above, of α-syn and HSF1 phosphorylation have detrimental effects in HD, whereas downstream effects of HTT phosphorylation are beneficial. Created with BioRender.com (accessed 29 June 2022).

**Figure 4 biomedicines-10-01979-f004:**
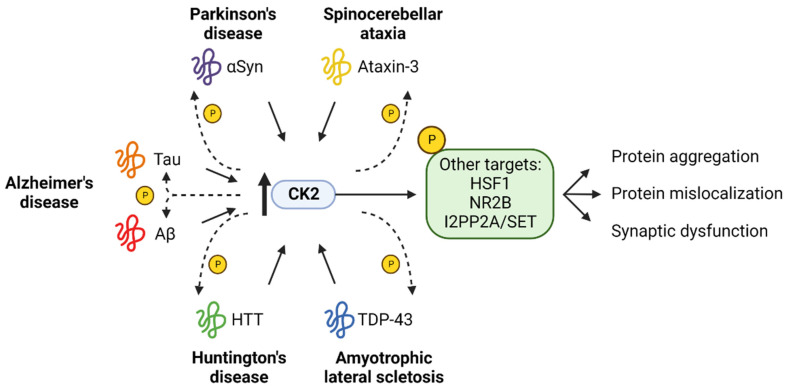
Activation of CK2 at the center of multiple neurodegenerative diseases. Upregulation of CK2 may serve as a convergent mechanism of neurodegeneration across several neurodegenerative disorders. CK2 mediates phosphorylation of the main pathological proteins for each of those diseases and further phosphorylates additional substrates. This working model proposes CK2-mediated phosphorylation of different targets influence disease pathology through mechanisms such as protein aggregation, protein mislocalization, and influencing synaptic dysfunction. Created with BioRender.com (accessed on 10 August 2022).

**Figure 5 biomedicines-10-01979-f005:**
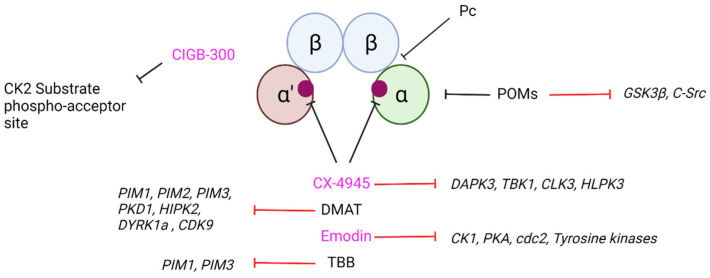
Summary of popular CK2 inhibitors, target sites and off-target effects. Several examples of available CK2 inhibitors with the sites of action are indicated in the figure. Inhibitors that have been used in or are currently in clinical trials are indicated with pink text, CX-4945, CIGB-300, and Emodin [147]. The example CK2 holoenzyme is a tetrameric form with 2-beta subunits (blue), one alpha subunit (green; right) and one alpha’ subunit (maroon; left). ATP binding sites are indicated by purple circles within the catalytic subunits. Off target effects of the inhibitors are indicated with red arrows [136,139,209,218,219,220,221]. Created with Biorender.com (accessed 29 June 2022).

**Table 1 biomedicines-10-01979-t001:** Palliative interventions used in HD treatment.

Agent	Target and Purpose	References
Citalopram (*Celexa*) Flouexetine (*Prozac*) Setraline (*Zoloft*)	Inhibitors of serotonin transporter SLC6A4 to treat depressive and aggressive symptoms	[42,43,44,45,46]
Venlafaxine (*Effexor*)	Sequential inhibitor of serotonin and norepinephrine reuptake transporters to treat depressive symptoms	[42,47]
Olanzapine (*Zyprexa*) Haloperidol (*Haldol*) Aripiprazole (Abilify) Risperidone (*Risperdal*)	D2 receptor antagonists used for the treatment of chorea	[42,48,49,50,51,52,53,54,55,56,57,58]
Clozapine (*Clozaril*)	Antagonist of the D1 and D4 dopamine receptors and 5-HT_2A_ and 5-HT_2C_ serotonin receptors used for treatment of chorea	[42,58,59,60,61,62]
Tetrabenzine (*Xenazine*)	Inhibits transport of monoamines via binding to type-2 vesicular monoamine transporters (VMAT2) used for the treatment of chorea	[37,42,58]
Rivastigmine (*Exelon*)	Inhibits activity of the cholinesterase enzymes, acetylcholinesterase (AChE) and butyrylcholinesterase (BuChE) used to treat cognitive deficits	[42,63,64]

## Data Availability

Not applicable.

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
