# Peer review of "Protein Kinase CK2 and Its Potential Role as a Therapeutic Target in Huntington’s Disease"

_biomedicines, 2022, doi:10.3390/biomedicines10081979_

Round 1

Reviewer 1 Report

The review untitled "Kinases and their potential as therapeutic targets in Huntington's disease" by White, McGlone and Gomez-Pastor describes a number of kinases that have been implicated at various levels in Huntington's disease (HD) and focusses largely on CK2, the Protein Kinase 2 also known as Casein Kinase 2 for historical reasons. The text is highly referenced with over 200 citations and the sections devoted to CK2 are very well detailed. 

My main comment is that the review could gain from being slightly more focussed. This could be easily achieved by centering the focus on CK2. Along those lines I thought that table 1 on antidepressants and antipsychotics (mostly targeting GPCRs and not kinases) was rather distracting. 

Because the world of kinases is so large, because so many kinases have been implicated directly or indirectly in so many neurodegenerative disease including Huntington's in a large number of assays (not always highly relevant), I believe that the quality of the review would increase is the focus was more clearly about CK2, starting with the title.

Following that logic, the paragraph related to other kinases (section 4) could be briefly introduced at the end of the review, if at all. If kept, it might help the readers to explain the logic of choosing those specific kinases (instead of: "include, but are not limited to") and not other important kinases such as GSK3, other CDKs, etc. With a focus on CK2 I do not think that this section is vital to the review. Finally, covering too many kinases so lightly might be misleading. For example the paragraph about CDK5 cannot stay as is. CDK5 has been involved in numerous biological functions and not just "regulator of neuronal migration in the developing central nervous system". The contradictory effects observed in mice and mentioned by the authors is obviously due to the many roles of CDK5. 

Instead of the large Table 1 rather out of place in this review, I would suggest to expand Figure 1, indicating all CK2 phospho-sites and putative ones as well, based on a general CK2 consensus for example (that could also be mentioned). This figure might be a good place to indicate non-CK2 kinases phospho-sites  as well and would further reduce the need for including 'other kinases' descriptions in the text. Because of numbering discrepancies in databanks, it might help the reader to identify phospho-positions by indicating on the figure the few amino acids around the phospho-site. 

The sections about PTMs that are not kinase related (3.1, 3.2, 3.3, 3.4) could be significantly reduced to focus on phosphorylation. Or the title should be more general perhaps? 

The section 6 is particularly important - and highlighting the role of CK2 in other neurodegenerative diseases as the authors do can only increase the importance and impact of CK2 therapeutic role. It might be worth considering adding a figure to summarize this section visually. Another suggestion could be to review that section, highlighting the medical importance of CK2, before focussing on HD. 

Author Response

We thank the reviewers for their constructive comments and criticisms. We have outlined a point-by-point response below to address the reviewers’ comments. We believe the suggested comments have significantly improve the review article. Reviewer’s comments are highlighted in Blue italics, and changes made to the manuscript were highlighted in track changes.

Reviewer 1

The review untitled "Kinases and their potential as therapeutic targets in Huntington's disease" by White, McGlone and Gomez-Pastor describes a number of kinases that have been implicated at various levels in Huntington's disease (HD) and focuses largely on CK2, the Protein Kinase 2 also known as Casein Kinase 2 for historical reasons. The text is highly referenced with over 200 citations and the sections devoted to CK2 are very well detailed. 

My main comment is that the review could gain from being slightly more focussed. This could be easily achieved by centering the focus on CK2. Along those lines I thought that table 1 on antidepressants and antipsychotics (mostly targeting GPCRs and not kinases) was rather distracting. 

We thank the reviewer for the constructive criticism to improve the focus of our review. We have simplified Table 1 by focusing on the molecular targets that current palliative treatments are targeting and by removing side effects and compressing drugs with the same molecular target. The new version of Table 1 now better fits with the review and points out the lack of treatments targeting kinases as the molecular target.

Because the world of kinases is so large, because so many kinases have been implicated directly or indirectly in so many neurodegenerative disease including Huntington's in a large number of assays (not always highly relevant), I believe that the quality of the review would increase is the focus was more clearly about CK2, starting with the title. Following that logic, the paragraph related to other kinases (section 4) could be briefly introduced at the end of the review, if at all. If kept, it might help the readers to explain the logic of choosing those specific kinases (instead of: "include, but are not limited to") and not other important kinases such as GSK3, other CDKs, etc. With a focus on CK2 I do not think that this section is vital to the review. Finally, covering too many kinases so lightly might be misleading. For example the paragraph about CDK5 cannot stay as is. CDK5 has been involved in numerous biological functions and not just "regulator of neuronal migration in the developing central nervous system". The contradictory effects observed in mice and mentioned by the authors is obviously due to the many roles of CDK5. 

We have modified the title of the manuscript from “Kinases and their potential as therapeutic targets in Huntington’s disease” to “Protein Kinase CK2 and its potential as a therapeutic target in Huntington’s disease”.

Regarding section 4, we decided to keep this section as we believe this section adds substance to the review by illustrating other kinases that hold merit to be studied in relation to HD. However, we have simplified this section describing other kinases involved in HTT phosphorylation and put more emphasis on the role of CK2 in HD.  We have added several statements to explicitly state the intent of that section and that it is not all encompassing of all kinases or all functions of those kinases (even within HD): “While these kinases are not the only kinases linked to HD and/or HTT phosphorylation, they were chosen as a representative examples to highlight the importance of studying kinases as potential therapeutic agents in HD.” (lines 260-262) & “A description of key kinases involved in HTT phosphorylation are detailed below:” (lines 267 & 268).

Additionally, we removed any mention of the general role of the kinases described to avoid misleading the readers and focus on the described role on HTT phosphorylation. We also added in the introductory paragraph the following statement: “However, while these kinases are linked to the phosphorylation of specific residues on HTT, the effects of targeting these kinases in the overall HD pathophysiology need careful interpretation since these kinases can phosphorylate a myriad of other targets.” (lines 262-265). In addition, at the end of the CDK5 section we also allude to contradictory effects potentially being due to other targets, “...the study suggests that other CDK5 targets, such as Src kinase, which impacts the localization of the NMDA receptor subunit NR2B, may play a more important role in disease progression than HTT phosphorylation.” (lines 316-318).

Instead of the large Table 1 rather out of place in this review, I would suggest to expand Figure 1, indicating all CK2 phospho-sites and putative ones as well, based on a general CK2 consensus for example (that could also be mentioned). This figure might be a good place to indicate non-CK2 kinases phospho-sites  as well and would further reduce the need for including 'other kinases' descriptions in the text. Because of numbering discrepancies in databanks, it might help the reader to identify phospho-positions by indicating on the figure the few amino acids around the phospho-site. 

We thank the reviewer for this suggestion. We have added a new Figure (now Figure 2) illustrating the different kinases responsible for mediating HTT phosphorylation as well as potential CK2 phospho-sites based on CK2 consensus sequences identified along HTT. We have also included amino acids around the phospho- or potential phospho- sites and have also indicated the NCBI sequence used to identify the potential sites. 

The sections about PTMs that are not kinase related (3.1, 3.2, 3.3, 3.4) could be significantly reduced to focus on phosphorylation. Or the title should be more general perhaps? 

To focus more on phosphorylation, we began the PTMs section discussing phosphorylation in detail and added a subsection ‘Other PTMs” to include a summary of sections 3.1, 3.2, 3.3 and 3.4 (line 213). We have also simplified the information provided in those sections not related to phosphorylation.

The section 6 is particularly important - and highlighting the role of CK2 in other neurodegenerative diseases as the authors do can only increase the importance and impact of CK2 therapeutic role. It might be worth considering adding a figure to summarize this section visually. Another suggestion could be to review that section, highlighting the medical importance of CK2, before focussing on HD. 

We thank the reviewer for these suggestions, and we have now included a new Figure (now Figure 4) to summarize the role of CK2 in other neurodegenerative diseases. The figure focuses on summarizing CK2 as a potential convergent mechanism in neurodegeneration across multiple neurodegenerative diseases.  

Reviewer 2 Report

Biomedicines-1820246

Type of manuscript: Review

Title: Kinases and their potential as therapeutic targets in Huntington’s disease

In this submitted manuscript, Angel White and colleagues reviewed the current state of knowledge in Huntington’s Disease pathology and the potential involvement of protein kinases in disease progression. The authors described current palliative therapies available and emphasised post-translational modifications of HTT protein and their involvement in toxicity. Authors described several kinases so far known to phosphorylate regulatory residues in HTT protein and their potential role in HD pathology and highlighted CK2 as a potential drug target in HD.

Overall, the review is nicely written and articulated rationally. I think this review will be of interest to researchers working in the HD field and also those researchers interested in studying molecular mechanisms of neurodegenerative diseases.

To improve the article further, I have the following minor suggestions.

1.      In Table 2: Please correct/update the citations (for Ser13) to match the rest of the table.

2.      Please correct typos (e.g., reviewed, patients) in lines 438, 441 and 442.

3.      In sub-heading 7 (CK2 inhibitors): Please list and describe all the inhibitors shown in Figure 3. To me, this is the critical information readers will get from this review.  E.g., what is Pc in the figure? Is it cyclo-peptide?

Author Response

We thank the reviewers for their constructive comments and criticisms. We have outlined a point-by-point response below to address the reviewers’ comments. We believe the suggested comments have significantly improve the review article. Reviewer’s comments are highlighted in Blue italics, and changes made to the manuscript were highlighted in track changes.

Reviewer 2

In this submitted manuscript, Angel White and colleagues reviewed the current state of knowledge in Huntington’s Disease pathology and the potential involvement of protein kinases in disease progression. The authors described current palliative therapies available and emphasised post-translational modifications of HTT protein and their involvement in toxicity. Authors described several kinases so far known to phosphorylate regulatory residues in HTT protein and their potential role in HD pathology and highlighted CK2 as a potential drug target in HD.

Overall, the review is nicely written and articulated rationally. I think this review will be of interest to researchers working in the HD field and also those researchers interested in studying molecular mechanisms of neurodegenerative diseases.

To improve the article further, I have the following minor suggestions.

  1. In Table 2: Please correct/update the citations (for Ser13) to match the rest of the table

We thank the reviewer for catching this mistake, the citations have been updated so they are all in the correct format.

  1. Please correct typos (e.g., reviewed, patients) in lines 438, 441 and 442.

Typos have been corrected

  1. In sub-heading 7 (CK2 inhibitors): Please list and describe all the inhibitors shown in Figure 3. To me, this is the critical information readers will get from this review.  E.g., what is Pc in the figure? Is it cyclo-peptide?

We have added general descriptions of each inhibitor to subheading 7; “i) 11-mer peptide (PC) [208] ii) polyoxometalates (POMs) aggregates of early-transition metal ions and oxo ligands [209], iii) CIGB-300 a peptide based inhibitor [210], iv) CX-4945 a small molecule ATP-competitive inhibitor [210], v) 2-dimethylamino-4,5,6,7-tetrabromo-1H-benzimidazole (DMAT) an ATP Competitive inhibitor is a Benzomidazole derivative [210] vi) Emodin is a Anthraquinone derivative naturally occurring in plants such as the Rheum palmate [210,211], vii) 4,5,6,7-tetrabromobenzotriazole (TBB) an ATP Competitive inhibitor is a Benzomidazole derivative [210].”  (lines 599-606)